# Surface Interactions during the Removal of Emerging Contaminants by Hydrochar-Based Adsorbents

**DOI:** 10.3390/molecules25092264

**Published:** 2020-05-11

**Authors:** Silvia Román, Joâo Manuel Valente Nabais, Beatriz Ledesma, Carlos Laginhas, Maria-Magdalena Titirici

**Affiliations:** 1Departmento de Física Aplicada, Universidad de Extremadura, Avda. Elvas s/n, CP: 06006 Badajoz, Spain; beatrizlc@unex.es; 2Departamento de Química, Escola de Ciências e Tecnologia, Universidade de Évora, Rua Romão Ramalho 59, 7000-671 Évora, Portugal; jvn@uevora.pt (J.M.V.N.); carloslaginhas@gmail.com (C.L.); 3Comprehensive Health Research Center (CHRC), NOVA Medical School, Faculdade de Ciências Médicas Campo Mártires da Pátria, 130, 1169-056 Lisboa, Portugal; 4Materials Research Institute, Queen Mary University London, Mile End Road, London E14NS, UK; m.m.titirici@qmul.ac.uk

**Keywords:** activated carbons, adsorption, hydrocarbonization, pharmaceutical effluents, water

## Abstract

The aim of this work was to test activated carbons derived from hydrochars produced from sunflower stem, olive stone and walnut shells, as adsorbents for emerging contaminants in aqueous solution, namely fluoxetine and nicotinic acid. The adsorption capacity was determined by the chemical nature of the adsorbents, namely the presence of specific functional groups and their positive or negative ionization in aqueous solutions and also by steric factors. The activated carbons produced by air showed a higher adsorption capacity of fluoxetine, whilst the samples produced by carbon dioxide activation were more useful to remove nicotinic acid. In general, surface acidity was advantageous for fluoxetine adsorption and detrimental for nicotinic acid removal. The adsorption mechanisms involved in each case were discussed and related to the adsorbents characteristics. The maximum adsorption capacity, Q_0_, given by the Langmuir model was 44.1 and 91.9 mg g^−1^ for fluoxetine and nicotinic acid adsorption, respectively.

## 1. Introduction

In recent years, the potential of renewable sources to either produce energy or added value materials has been a challenge in all fields of research. Focusing on biomass waste, its conversion into valuable products is particularly interesting, especially if the process allowing its transformation can be classified as so-called green chemistry.

Studies have demonstrated that the hydrothermal carbonization (HTC) of biomass results in the formation of beneficial materials without entailing greenhouse gas emissions. This process is advantageous in relation to other thermochemical processes such as pyrolysis, due to its simplicity and very low operating costs, in addition to the fact that it allows high solid yields to be obtained, and retains the most of the biomass carbon on the final product [1].

HTC can be defined as a process in which a substrate is placed in water and heated at moderate temperature (150–350 °C) in a closed system under autogenous pressure conditions. As a consequence, a carbon-rich material (hydrochar, HC) is obtained. Due to its enhanced carbon content, the HC can be potentially utilized in several applications. Among them, their use as energy carriers has proven to be very effective since these materials often have heating values which make them competitive with other carbon fuels such as lignite or bituminous coal [1,2]. Other pieces of research have investigated the use of these materials for gas storage applications [3]. Regarding their use as adsorbents, the majority of the research points out the low porosity of the HCs, likely due to incomplete carbonization and the permanence of disorganized matter within the porous network [1,4,5,6]. In order to increase the porous structure as well as create specific functionalities, different studies on activation of HCs have been carried out. The potential of physical and chemical activation methods to produce adsorbents is of high interest due to the low cost associated with the non-use of high temperature pyrolysis processes.

Most of the research carried out on this topic reports the activation by KOH [7], H_3_PO_4_ [8], and lithium chloride [9]. Additionally, a few studies were published about the physical activation of HCs [10]. Román et al. [11] have prepared, in a previous work, ACs from HCs derived from various biomass wastes, namely sunflower steam, walnut shell and olive stone, by means of physical activation with air and carbon dioxide. Due to the differences between each activating agent, as well as the temperatures employed in each case, it was possible to tune not only the porosity characteristics but also the chemical surface features of the HCs. The samples produced in reference [11] were used in the work now reported.

The relevant role of surface chemistry on adsorption processes is well documented [12,13]. The prominence of acid or basic functionalities of an adsorbent can enhance the adsorption selectivity towards a given adsorbate, via the participation of non-dispersive specific interactions as well as electrostatic ones. There is some evidence that in particular cases, porosity characteristics of adsorbents have a negligible influence on the adsorption process, and it is the chemical surface which brings under control the adsorption capability of adsorbents [14].

Concern has been expressed regarding the presence of pharmaceuticals (that are increasingly consumed in developed countries [15]) in the environment. Ground waters are the final deposit of these compounds, due to their low volatility and low reactivity, and have multiple effects on ecosystems that are not well known yet, especially in the long term [16]. For their removal, adsorption onto ACs is regarded as a consolidated and efficient process, providing many advantages in comparison with other methods, such as low cost, green character and ease of implementation.

As a further step to evaluate their application as adsorbents, this study aimed to analyze the capability of the activated HCs described in [11] as adsorbents of two emergent contaminants, namely fluoxetine and nicotinic acid removal, in aqueous solution, under neutral pH conditions. Both compounds have received attention in recent years. On the one hand, fluoxetine (IUPAC name: n-methyl-3phenyl-3-[4-(trifluoromethyl)phenoxy]propan-1-amine hydrochloride) is a selective serotonin reuptake inhibitor drugs, which is currently used to treat a variety of major psychiatric pathologies such as depression, eating disorders, anxiety, and obsessive-compulsive behavior [17]. Despite its abundancy of aquatic streams, only few pieces of research have studied the removal of fluoxetine by adsorption, and most of them have used commercial ACs, alone or combined with carbon nanotubes [18]. In the frame of biomass or waste based adsorbents, the recent work of Silva et al. [19] can be highlighted, where the biosorption of fluoxetine on several biomass sources (spent coffee grounds, pine bark and cork waste) was studied.

On the other hand, nicotinic acid (IUPAC name: pyridine-3-carboxylic acid, commonly named Vitamin B3) is used as a medicine to decrease the risk of anemia and cardiovascular events [20], although it has also been associated with carcinogenic effects. Very few studies have been devoted to the investigation of nicotinic acid removal by adsorption processes onto commercial ACs [21], and, to date, none of them have used biomass-based adsorbents.

To the best of the authors’ knowledge, the removal of none of these adsorbates has been studied before by adsorption onto HC-based ACs. These materials could be good candidates for these processes since, apart from suitable porosity features, they offer a wide range of surface functionalities that might favor specific interactions. Provided the structural differences between these two compounds, this study was aimed at elucidating how the dissimilar surface characteristics of the adsorbents might influence their adsorption process mechanisms.

## 2. Results and Discussion

### 2.1. Fluoxetine Adsorption

The fluoxetine adsorption isotherms onto the HC-derived ACs can be seen in Figure 1. All isotherms can be classified as L-type, according to Giles classification [22]; they present an initial rise in the low equilibrium concentration region followed by a plateau. It is noteworthy that the same pattern was obtained for fluoxetine adsorption but using activated carbons prepared by chemical activation with KOH [23] or by physical CO_2_ and steam activation of almond tree pruning [24].

The activated samples produced by carbon dioxide activation, in general, present a more abrupt slope denoting a greater affinity at lower concentrations, while air-activated ones show a slower rise to the plateau. These results indicate that although the enhanced pore volumes of samples WSC, SFC and OSC might be beneficial for adsorption at low concentrations, the determinant factor affecting fluoxetine adsorption capability at higher values of concentration is rather related to surface chemistry. In particular, the presence of oxygen functionalities and hence, a greater surface acidity leads to a better adsorption performance of samples WSA, SFA and OSA. This effect can be related to the chemical structure of fluoxetine (Figure 2), with the presence of the strong dipole in HCl and NH_2_. This can favor specific interactions with electron-donating surface groups of the carbons such as OH and COOH, which are more abundant for the air activated samples [11].

Another plausible contribution to the adsorption mechanism are the dispersive π–π coupling reactions, which can take part in any adsorption system involving aromatic compounds. These interactions are more effective in the case of basic adsorbents, whose basal plane electrons are more delocalized [25]; in consequence, CO_2_ activated carbons would be more likely to undergo these adsorption mechanisms.

On the other hand, the presence of electrons on nitrogen and chlorine atoms produces dipolar moments for fluoxetine. Therefore, negative charges are close to these atoms and the presence of the polar oxygen groups on the carbon surface with lone pairs of electrons on their oxygen atoms may be the reason for the surface-specific interactions between the oxygen surface groups of carbon samples and fluoxetine molecules.

Finally, it has to be taken into account that under the experimental conditions used, i.e., neutral pH, both the adsorbate and adsorbent show a specific charge. Considering the speciation diagram and pk_a_ value, it can observe that fluoxetine is predominantly on its protonated form [24].

Regarding the ACs, the ionization of the surface functional groups and thus the electric mean charge of the material surface depend on the difference between the solution pH and the adsorbent PZC. For acid adsorbents, when the pH value of the solution is smaller than the PZC value of the carbon, a net negative charge is developed, as it happens in samples WSA, OSA and SFA. The carbons with basic characteristics (WSC, OSC and SFC) have a predominance of positive charges as their value of PZC is smaller than the solution pH. The positive charge of air activated samples therefore favor electrostatic attractive forces. In Figure 2, the cited different contributions to the adsorption mechanism are illustrated.

The fitting of the experimental data by the models Langmuir, Freundlich, Redlich–Peterson and Sips allow us to better describe the adsorption. The estimated parameters for each model are listed in Table 1; in all cases, we used the non-linear equation of the models to avoid interferences due to the modification of error distribution, as a consequence of linearization processes [26]. The model that provided the best fit was the Langmuir model; thus, it was the model that was used for modelling the data on Figure 1.

The values of Q_0_ provided by Langmuir model were in the range of 9.9–44.1 mg g^−1^. These values were greater for air-activated samples, with the exception of walnut shell, we can observe a significant similarity between WSA and WSC. The value of OSA was outstanding, which may be related with its greatest acidity, although this sample was also the one with the worst fitting coefficient. Estimated adsorption capacities were, respectively, similar and higher than those reported for ACs chemically activated by ZnCl_2_ and NaOH [23]. Likewise, these obtained Q_0_ values are slightly lower than those obtained for almond pruning ACs obtained by physical activation [24]. However, it is worth mentioning that the cited papers all delt with the use of ACs which were prepared by traditional activation techniques, which involve a pyrolytic stage. In comparison with HTC, the environmental, energetic and economical costs of pyrolysis are much higher, which confers an additional value to the present adsorbents. In addition, it has to be highlighted that the ACs produced in this work show a better performance towards fluoxetine than biosorbents prepared from biomass wastes as those reported in [19], which attained Q_o_ values in the range of 6.4–14.3 mg g^−1^.

The values of the affinity constant, K_L_, are greater for carbon dioxide ACs. This might be related with the slightly smaller pore size of these adsorbents. Their lower hydrophilicity, as suggested by its less abundance of oxygen functionalities, can also explain this behavior. For lower values of C_e_, competitive adsorption with water molecules might explain the slower attainment of the plateau in the case of air ACs.

With respect to Freundlich model, it is worth mentioning that the model fitting was better in the case of air activated samples. This might be related to the fact that the Freundlich isotherm provides a continuous increase of *q_e_* in a wide range of *C_e_* values; in fact, one of the weaknesses of this approach is related to this effect, as a finite limit is not reached. Possibly, this is the reason why the adjustment of samples activated by carbon dioxide is worse, since the plateau in these carbons is better defined than air ones.

Likewise, the values of n are not very close to zero, suggesting a low surface heterogeneity and are of the same order of magnitude than those reported in previous works [26]. Moreover, n is greater than 1 in all cases, which indicates that adsorption is favorable. Finally, *K_F_* values are lower for air ACs, in coherence with their slower attainment of the plateau.

The worst adjustment of the data corresponded to the Redlich–Peterson model. In accordance with previous models, the constant *a_R_*, related to adsorption intensity, was slightly greater for CO_2_ ACs, as it happened in the case of parameter *K_L_* from Langmuir equation. Moreover, the high value of *K_R_* for OSA was outstanding, in coherence with the largest fluoxetine uptake of this carbon. Regarding th eSips model, it predicted quite well the adsorption behavior, with the exception of the OSC sample (R^2^ = 0.81). The values of α_S_—associated with the maximum adsorption capacity—were rather similar to those featured by the isotherms. Likewise, the values of *K_s_* and β_S_ did not exhibit significant differences, although both were slightly greater for CO_2_ ACs, consistently with a larger isotherm slope at low values of equilibrium concentration.

### 2.2. Nicotinic Acid Adsorption

Nicotinic acid adsorption isotherms are plotted in Figure 3. Like for fluoxetine adsorption, the curves can be considered L-type, which indicates a significant affinity of the carbons for the adsorbate. In the case of WSC and SFC we can observe a higher adsorption capacity than the WCA and SFA, respectively. Dissimilarly, in the case of olive stone samples (OS), the activating agent had no significant influence on the nicotinic acid adsorption.

The differences on the surface chemistry of the adsorbents are possibly a decisive factor to explain these results. On the one hand, the enhanced basicity of the carbon dioxide-activated carbons provides a greater delocalization of the basal plane electrons [25] so that these adsorbents are more prone to undergo π–π interactions with the nicotinic acid aromatic ring. On the other hand, the existence of dipoles on the adsorbate has to be taken into account, since they can participate in specific chemical interactions. Firstly, the nitrogen included on the aromatic structure has a tendency to withdraw electrons from the aromatic ring; secondly, a negative charge is displaced towards the oxygen in the double bond of carboxylic acid. Both charge distributions can be responsible for dipolar interactions with the carbons surface functionalities. The electrostatic interactions cannot be undervalued and might be of great prominence in the system. In aqueous solution, nicotinic acid is dissociated into its negative form at a pH greater than its pk_a_, which is 2.7 for the COOH group and 4.74 for the ring nitrogen [27]. From Table 2 (shown in materials section, where the adsorbents used are characterized), it can be observed that activated carbons produced by carbon dioxide activation are all basic and in consequence, since their PZC is greater than the solution pH, have a net positive charge. This, in turn, would favor the electrostatic attractive interactions between the adsorbent and the anion, which is in line with a greater adsorption capacity shown by these activated carbons. The proposed adsorption mechanisms are in line with that proposed by Ayranci and Duman [28], which investigated the adsorption of nicotinic acid onto high area commercial activated cloths. In Figure 4, the mentioned mechanism of the adsorption is illustrated.

Likewise, regarding fluoxetine adsorption, Langmuir model was the one that provided the best fitting for the nicotinic acid adsorption; as can be seen in Table 2, all samples except OSA show a R^2^ superior to 0.98. The fitting curves in Figure 3 were obtained by the application of the Langmuir model to the experimental data.

First of all, it is visible that in all cases, the ACs tested have a higher adsorption capacity for nicotinic acid than for fluoxetine. This might be related to the lower size of the former molecule26 [26], allowing it to have an easier access to the adsorption sites. Maximum adsorption capacities (Q_0_) in the range of 15.2–97.8 mg g^−1^ were obtained. These values are quite larger than those obtained by Datta [21], Ayrancy and Dumas [28], and Qureshi et al. [29], who used commercial adsorbents with high values of surface area.

The adsorption features deduced from the shape of the adsorption isotherms are well described by the tendencies found in the adsorption parameters estimated by the models applied. For example, for the Langmuir model, *K_L_* values are in general lower for air-activated carbons, in coherence with the slower attainment of the plateau for these samples. This parameter is particularly high for a sample SFC.

The Freundlich model provided *K_F_* values that are, in all cases, larger than those found in the case of fluoxetine, in accordance with the greater isotherm rise at low relative concentrations. Sample SFA can be highlighted because of its worst adjustment, which might be associated to the fact that these isotherms exhibit a well-defined plateau, and therefore, is better defined by Langmuir model. Likewise, the low Freundlich exponent (*n* value) of OSA is remarkable, due to the low adsorption intensity at low *C_e_*. Also, WSA stands out due to its low adsorption capacity, also reflected by its small value of *K_F_*. Finally, the Redlich–Peterson and Sips models also illustrated quite well the results of nicotinic acid. In both cases, the values of the parameters related to adsorption capacity, followed the trends already described for Langmuir and Freundlich models, in accordance with the isotherms.

As final remark, we would like to emphasize how, once the preparation of the adsorbents is optimized (in terms of control of porous structure and surface functionality), testing them is essential to investigate their performance towards each adsorbate. Two organic drugs, such as fluoxetine and nicotinic acid, can show enhanced affinity towards one adsorbent that, at first glance, could seem less interesting. For this reason, taking the specific surface or the pore volumes is not enough, because the pore size distribution and the amount and type of functional groups can be decisive to favor adsorption or repulsion processes. The authors would also add that real adsorption in wastewater plants takes place with multiple adsorbates, and, in this line, future steps will be devoted to studying the performance of these type of materials in multi-sorption systems, mimicking the conditions of real cases.

## 3. Materials and Methods

### 3.1. Materials

Fluoxetine hydrochloride (reference standard) was kindly provided by Eli Lilly (Indianapolis, IN, USA), while Nicotinic acid (98% purity) was supplied by Sigma Aldrich (St Louis, MO, USA).

The adsorbents used in this study were all made by HTC of three biomass materials (walnut shell, sunflower stem, and olive stone), followed by physical activation. The precursors were chosen because of their abundancy in the Iberian Peninsula, and were supplied by local cooperatives from Extremadura (walnut shell and olive stone) and Alentejo (sunflower stem) areas, in Spain and Portugal, respectively. All the three materials were used “as received” (without any washing), and then subjected to crushing and grinding to a particle size of 1–2 mm, as described in previous papers [2].

The HTC preparation conditions were 200 °C, for 20 h, and ratio biomass/water of 5%, and the reactions were performed into a 0.15 L autoclave (Berghof, Germany). Each HC was then activated on a vertical stainless steel furnace following two different methods (a more detailed description can be found in [11]): a) air activation processes, using a flow rate of 100 mL min^−1^, dwell time of 30 min and temperature of 250 °C, and b) carbon dioxide (40 mL min^−1^) at 850 °C, during 30 min. A N_2_ flow (100 mL min^−1^) was fed to the system during the heating and cooling periods.

Six samples were prepared and named XY, where X denotes the precursor (WS: walnut shell; SF: sunflower; and OS: olive stone), and Y refers to the activating agent (A: air; and C: carbon dioxide). Table 2 includes the main textural parameters as well as the point of zero charge (PZC), data already published in [11]. The porosity parameter was obtained according to suitable models and the PZC was determined by titration, as described in the cited work.

It is interesting to recall the significant increase in S_BET_ that the activation brought out, in relation to the original HCs, for which it had values of 31, 27 and 22 m^2^ g^−1^, respectively. Moreover, all the three HCs were very acidic in nature before the activation processes (4.12, 4.01 and 3.95, following the same sequence).

On the other hand, it is also interesting to notice that while CO_2_ processes resulted in a similar porosity development (in all cases microporous carbons with S_BET_ values close to 400 m^2^ g^−1^), air behaved better with sunflower stem. Based on previous research with the same precursors on traditional air activation processes, the authors believe that, in fact, walnut shell and olive stone did not develop all their potential porosity [30,31]. In this work, the results of high burn-off for these two materials after air activation and also the bigger contribution of mesoporosity suggest that some external burning might have taken place.

### 3.2. Adsorption Isotherms

Fluoxetine and Nicotinic Acid adsorption isotherms were determined using a batch analysis, 0.010 g of adsorbent was added to 15 mL of aqueous solution of each contaminant with initial concentrations ranging from 10 to 200 mg L^−1^, and the solutions were made with ultrapure water at neutral non-fixed pH. The flasks were then placed in a thermostatic bath at 298 K and allowed to equilibrate for 48 h, since previous experimentation on the adsorption kinetics showed that this period of time was enough to guarantee equilibrium [24,32].

After equilibration, the adsorbents were filtered, and the concentration of each substance in the supernatant solutions analyzed by UV/Vis spectrophotometry (spectrophotometer UNICAM Helios-λ) at a wavelength of 274 and 262 nm, for fluoxetine and nicotinic acid, respectively. This wavelength was selected after previous spectral scanning tests, which showed the stability of this signal (λ_max_), independently of the possible pH variations. Each measurement was made in triplicate and the absorbance used to calculate the adsorbate concentration corresponds to the average value.

### 3.3. Adsorption Models

In this study, various adsorption models were applied for experimental data fitting, namely two-parameters models (Langmuir and Freundlich) and three-parameters models (Redlich-Peterson and SIPs). These models can provide an extra set of parameters which complement the information depicted from the experimental isotherms. In the case of three parameters models, the equations were solved by maximizing the correlation coefficient between the experimental data points and theoretical model predictions with the solver add-in function of Microsoft Excel. The equations describing the models used are given below:
(a)The Langmuir model:*q_e_* = (*Q_0_C_e_K_L_*)/(1 + *C_e_K_L_*)(1)
where *q_e_* (mg g^−1^) is the measured adsorption at a drug equilibrium concentration of *C_e_*. *Q_0_* (mg g^−1^) is the maximum adsorption capacity of the monolayer and *K_L_* (L mg^−1^) is the Langmuir constant, related to the free energy of adsorption.(b)The Freundlich model:*q_e_* = *K_F_C_e_*^1/n^(2)
where *K_F_* (mg g^−1^)(L mg^−1^)^1/n^ is the Freundlich constant representing the adsorption capacity and *n* (dimensionless) is the constant depicting the adsorption intensity.(c)The Redlich–Peterson model [33] is an empirical isotherm incorporating three parameters (Equation (3)). It combines elements from the Langmuir and Freundlich equation, and the mechanism of adsorption does not follow ideal monolayer adsorption.
*q_e_* = (*K_R_C_e_*)/(1 + *a_R_C_e_^g^*)(3)
where *K_R_* is an isotherm constant (L g^−1^), a_R_ is the Redlich–Peterson isotherm parameter (L mg^−1^), and *g* is the Redlich–Peterson isotherm exponent.(d)The SIPs model [34] uses an equation similar to the Freundlich equation, but it has a finite limit when the concentration is sufficiently high; in this way, this model avoids the continuous increase in the adsorbed amount with an increasing concentration.
*q_e_* = (*K_S_C_e_^βS^*)/(1 + α*_S_C_e_^βS^*)(4)
where *α_S_* is the Sips maximum adsorption capacity (mg g^−1^), *K**_S_* the Sips equilibrium constant (L mg^−1^), and *β_S_* the Sips model exponent.

## 4. Conclusions

The activation of biomass-based hydrochars using air and carbon dioxide can provide adsorbents with diverse porosity and surface acidity properties that can be used to remove fluoxetine and nicotinic acid, although the most suitable option is strongly dependent on the adsorbate.

It was found that adsorption process was more influenced by the surface chemistry than by the porous structure of the carbons. Fluoxetine showed removal results when air activated carbon were used, whilst in the case of nicotinic acid, the maximum adsorption capacity was achieved for samples produced by carbon dioxide. This behavior can be justified by the non-dispersive interaction, chemical bonding and electrostatic interactions between the activated carbons surface and the adsorbate species.

Among the different adsorption models used (Langmuir, Freundlich, Redlich-Peterson and Sips), the first one predicted more accurately liquid the adsorption isotherms. Values of Q_0_ up to 44.1 and 91.9 mg g^−1^ were obtained for fluoxetine and nicotinic acid adsorption which were, respectively, similar or superior to those found in the literature for other commercial and home-made adsorbent. The convenience of using hydrocarbonization followed by activation is noteworthy because of its simplicity, low energetic and economical cost and environmental advantages.

## Figures and Tables

**Figure 1 molecules-25-02264-f001:**
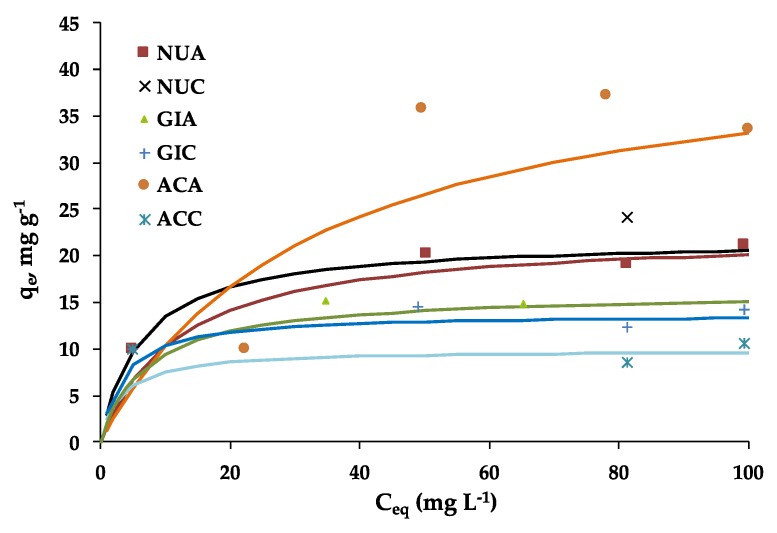
Fluoxetine adsorption isotherms.

**Figure 2 molecules-25-02264-f002:**
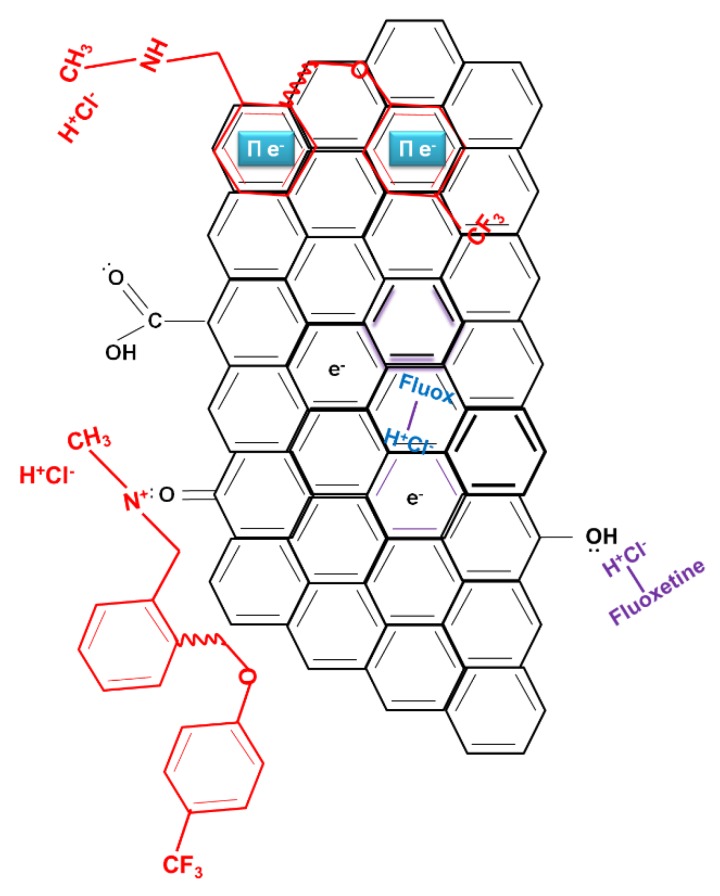
Mechanisms of adsorption of fluoxetine onto activated hydrochars.

**Figure 3 molecules-25-02264-f003:**
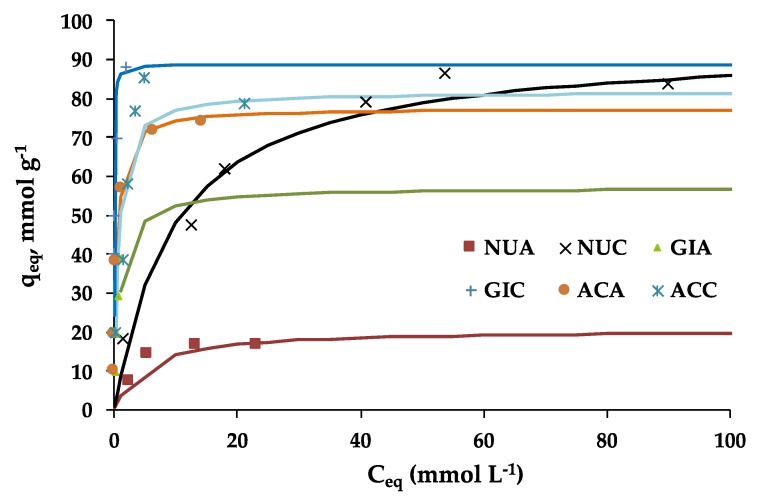
Nicotinic acid adsorption isotherms.

**Figure 4 molecules-25-02264-f004:**
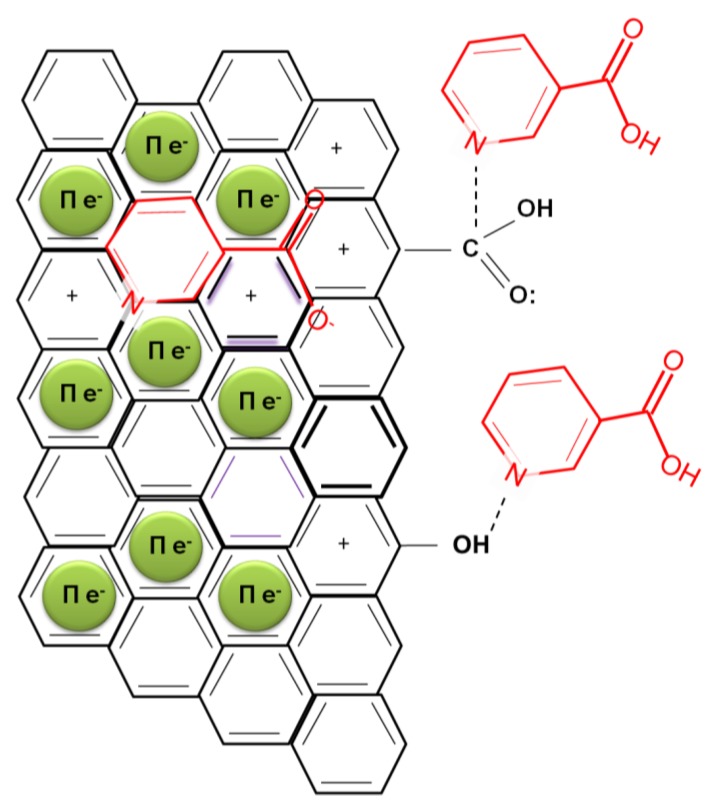
Mechanisms of adsorption of nicotinic acid onto activated hydrochars.

**Table 1 molecules-25-02264-t001:** Model parameters and correlation coefficients.

**Fluoxetine**	**Parameters**	**SFA**	**SFC**	**OSA**	**OSC**	**WSA**	**WSC**
Langmuir	q_0_ (mg g^−1^)	16.05	13.77	44.07	9.87	22.43	26.26
	K_L_ (L mg^−1^)	0.14	0.77	0.03	0.44	0.07	0.40
	R^2^	1.00	0.98	0.95	0.99	0.99	0.98
Freundlich	K_F_ (mg g^−1^)(L mg^−1^)^1/n^	6.10	8.59	2.28	5.06	0.26	9.70
	n	3.02	9.35	1.62	15.40	4.26	4.72
	R^2^	0.97	0.71	0.95	0.85	0.97	0.75
Redlich-Peterson	K_R_ (L g^−1^)	23.00	32.00	72.40	25.70	26.02	20.10
	a_R_ (L mg^−1^)	1.45	2.10	1.24	3.12	2.41	0.82
	g	0.91	1.00	0.98	0.98	0.91	1.03
	R^2^	0.88	0.75	0.93	0.83	0.78	0.78
Sips	K_S_ (L mg^−1^)	0.66	0.77	0.85	0.94	0.76	0.88
	α_S_ (mg g^−1^)	16.20	14.20	34.50	9.55	21.20	24.50
	β_S_	0.85	1.35	0.46	0.68	0.59	0.71
	R^2^	0.96	0.96	0.93	0.81	0.96	0.96
**Nicotinic Acid**	**Parameters**	**SFA**	**SFC**	**OSA**	**OSC**	**WSA**	**WSC**
Langmuir	q_0_ (mg g^−1^)	57.21	89.59	77.44	81.93	16.18	91.91
	K_L_ (L mg^−1^)	1.14	4.41	2.42	1.64	0.86	0.12
	R^2^	1.000	0.999	1.000	0.814	0.999	0.992
Freundlich	K_F_ (mg g^−1^)(L mg^−1^)^1/n^	20.42	66.31	21.68	41.53	5.77	17.43
	n	4.49	2.90	7.25	4.09	3.82	2.58
	R^2^	0.76	0.98	0.98	0.84	0.89	0.96
Redlich-Peterson	K_R_ (L g^−1^)	28.90	286.55	212.00	224.50	26.02	30.61
	a_R_ (L mg^−1^)	0.49	3.12	2.33	3.60	2.13	1.14
	g	1.00	0.99	0.65	0.85	0.99	0.69
	R^2^	0.81	0.96	0.92	0.74	0.94	0.99
Sips	K_S_ (L mg^−1^)	4.20	4.35	3.90	4.20	4.70	4.15
	α_S_ (mg g^−1^)	52.00	89.20	75.50	78.60	13.70	88.50
	β_S_	1.60	1.75	1.03	1.98	0.36	0.18
	R^2^	0.99	0.99	0.89	0.82	0.92	0.94

**Table 2 molecules-25-02264-t002:** Textural parameters and point of zero charge of the adsorbents [11].

Samples	S_BET_ m^2^ g^−1^	V_mi_ cm^3^ g^−1^	V_me_ cm^3^ g^−1^	V_ma_ cm^3^ g^−1^	PZC
WSA	213	0.105	0.052	2.361	4.43
WSC	379	0.196	0.017	2.253	8.53
SFA	434	0.228	0.031	6.292	4.25
SFC	438	0.230	0.047	5.211	8.12
OSA	204	0.115	0.002	2.094	4.05
OSC	438	0.231	0.006	3.558	9.46

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
