# Peer review of "Surface Interactions during the Removal of Emerging Contaminants by Hydrochar-Based Adsorbents"

_molecules, 2020, doi:10.3390/molecules25092264_

Round 1
Reviewer 1 Report
Surface interactions during the removal of emerging
contaminants by hydrochar based adsorbents
There are some minor issues especially regarding the methodology that must be fixed before the manuscript can be accepted for publication. The corrections needed are minor and will improve the manuscript.
Below is some detailed comments
Introduction
- The introduction is rather well written even if the number of references is rather low. Most of the statements are supported with only one reference. Authors are strongly suggested to add more references.
2.1. Materials
- The activation process must be better described, Temperatures used, temperature ramps, atmosphere etc.
- The data presented in Table 1, please describe how it was obtained. Degassing parameters, adsorption parameters. Also the calculations made to obtain the results and models used must be described.
Results
- Even if some results have been published earlier regarding the adsorption time it would be nice if the authors could show that there is a real equilibrium after 48 hours = at least 5 points on the flat part of the curve . Authors are strongly suggested to add this data.
- The lack of the equilibrium data makes the following data and figures rather unreliable.
- The mechanisms presented in Figure 4 is rather interesting. Sorry to say there is no data presented in the manuscript that supports the presence of the structures presented in Figure 2 and Figure 4. Up to me speculations are presented as data. Please supply some more evidence for the presence of the structures presented, maybe XPS or IR?
Author Response
Answers to reviewer 1,
The first thing we would like to thank the reviewer because, even during this confinement period, found the time to review our work and make it better. We are very glad that he/she liked our work and we hope that after all the changes we have made, he/she like it more, ;). Below we explain point by point how we have followed the suggestions of the reviewers:
The introduction is rather well written even if the number of references is rather low. Most of the statements are supported with only one reference. Authors are strongly suggested to add more references.
We understand that updating and enlarging the references was very convenient, and, in this way, we have reviewed this part of the manuscript, adding several references. We have also modified the redaction and improved the description of the main target of the work.
The new references we have added to the introduction have been:
[15] Health at a glance 2018: State of health in the eu cycle. European Comission Report. 2018.
[16] Li, Y.; Zhang, S.; Zhang, W., Xiong, W.; Ye, Q.; Hou, X.; Wang, C. Wang, P. Life cycle assessment of advanced wastewater treatment processes: Involving 126 pharmaceuticals and personal care products in life cycle inventory. J Env Manag 2019, 238, 442–450.
[18] Sousa-Moura, D.; E. Y. Matsubara et al. CNTs coated charcoal as a hybrid composite material: Adsorption of fluoxetine probeby zebrafish embryos and its potential for environmental remediation. Chemosphere, 2019, 230, 369-376.
[19] Silva, B.; Martins, M.; Rosca, M.; Rocha, V.; Lagoa, A.; Nevesa, I. C.; Tavares, T. Waste-based biosorbents as cost-effective alternatives to commercial adsorbents for the retention of fluoxetine from water. Sep Purif 2020, 235, 116-139.
[20] Bruckert, E.; Labreuche, J.; Amarenco, P. Meta-analysis of the effect of nicotinic acid alone or in combination on cardiovascular events and atherosclerosis. Atherosclerosis 2010, 210(2), 353–361.
[21] Datta, D.; Sah, S.; Rawat, N.; Kumar, R. Application of Magnetically Activated Carbon for the Separation of Nicotinic Acid from Aqueous Solution. J Chem Eng Data 2017, 62, 712−719.
The activation process must be better described, Temperatures used, temperature ramps, atmosphere etc.
Although this procedure has been previously described in a previous work, we have enlarged a little this part of the manuscript.
“Each HC was then activated on a vertical stainless steel furnace following two different methods (a more detailed description can be found at ref. [11]): a) air activation processes, using a flow rate of 100 mL min-1, dwell time of 30 min and temperature of 250ºC, and b) carbon dioxide (40 mL min-1) at 850 ºC, during 30 min. A N2 flow (100 mL min-1) was fed to the system during heating and cooling periods.”
The data presented in Table 1, please describe how it was obtained. Degassing parameters, adsorption parameters. Also the calculations made to obtain the results and models used must be described.
Since the paper is based on the use of adsorbents prepared in a previous work, which is cited several times in the text, we think that including this type of details is redundant. In order to clarify this information we have added the following sentence to the revised version of the manuscript:
“The porosity parameter were obtained according to suitable models and the PZC was determined by titration, as described in the cited work.”
Even if some results have been published earlier regarding the adsorption time it would be nice if the authors could show that there is a real equilibrium after 48 hours, at least 5 points on the flat part of the curve. Authors are strongly suggested to add this data.The lack of the equilibrium data makes the following data and figures rather unreliable.
Unfortunately, we did not study the kinetics of adsorption; we can make sure that after 48 h, the equilibrium had been reached, because we have measured the concentration after this time period and checked that it did not vary, but we did not study how the concentration of the adsorbents changed with time.
If, after this explanation, the reviewer considers that we should test the original biomass materials and the HCs, it would involve additional experimental work that in addition we are not in conditions of doing (the University is closed due to the confinement). We hope the reviewer and the editor accept out point of view.
The mechanisms presented in Figure 4 is rather interesting. Sorry to say there is no data presented in the manuscript that supports the presence of the structures presented in Figure 2 and Figure 4. Up to me speculations are presented as data. Please supply some more evidence for the presence of the structures presented, maybe XPS or IR?
As previously indicated we presented the spectra of the different HCs in a previous work (ref. 11); we consider that including this information again is redundant. Besides, in our discussion we are mostly based on the surface PZC and the property of “mostly basic” or “mostly acid” (assuming that ACs are amphoteric in nature). In this way, and according to the approach suggested by Castillo et al., we associate a greater acidity with a lower delocalization of electrons and viceversa. Also, acid adsorbents are associated to the migration of protons to water, and thus a final negative charge (and the opposite trend for basic carbons).

Reviewer 2 Report
In this manuscript, the authors studied the potential application of six types of lignocellulosic biomass based hydrochars regarding adsorption of two contaminants namely fluoxetine and nicotinic acid from aqueous solution, as an extended research to their former published work. Overall, the topic of the manuscript falls into the journal scope and the experimental data are solid, which might be beneficial to the readers. I suggest a minor revision prior to the possible publication in this journal. Remarks and suggestions are shown underneath:
1. Please carefully review the English language through the manuscript as there are mild grammatical errors and typos can be still found in the manuscript. e.g. the addressed affiliation d seems to be c.
2. Improvements can be made to the introduction section to be more insightful. Moreover, most of the listed references were published before 2015 while some of which even earlier. Please read these newly published related articles which might offer references to retrieve useful information regarding the adsorption process as well.
1) Li, Z., Sellaoui, L., Dotto, G. L., Bonilla-Petriciolet, A., & Lamine, A. B. (2019). Understanding the adsorption mechanism of phenol and 2-nitrophenol on a biopolymer-based biochar in single and binary systems via advanced modeling analysis. Chemical Engineering Journal, 371, 1-6.
2) Li, Z., Hanafy, H., Zhang, L., Sellaoui, L., Netto, M. S., Oliveira, M. L., ... & Li, Q. (2020). Adsorption of congo red and methylene blue dyes on an ashitaba waste and a walnut shell-based activated carbon from aqueous solutions: Experiments, characterization and physical interpretations. Chemical Engineering Journal, 388, 124263.
3) Li, Z., Wang, G., Zhai, K., He, C., Li, Q., & Guo, P. (2018). Methylene blue adsorption from aqueous solution by loofah sponge-based porous carbons. Colloids and Surfaces A: Physicochemical and Engineering Aspects, 538, 28-35.
3. The conclusion seems somewhat lengthy and mild concentration is suggested.
Author Response
Answers to reviewer 2,
The first thing we would like to do is to thank the reviewer for his/her time and effort to improve our manuscript; we are really motivated that he/she liked the work and we are delighted to follow his/her suggestions. Specially in the frame of this strange situation we are living, he/she found the time to devote his/her knowledge to this work, and we appreciate it very much. We have answered the suggestions point by point, as we explain below:
1) 1. Please carefully review the English language through the manuscript as there are mild grammatical errors and typos can be still found in the manuscript. e.g. the addressed affiliation d seems to be c.
We have carefully reviewed the English style and corrected several grammar and vocabulary mistakes. We have also rewritten several paragraphs to help the understanding of different concepts.
- Improvements can be made to the introduction section to be more insightful. Moreover, most of the listed references were published before 2015 while some of which even earlier. Please read these newly published related articles which might offer references to retrieve useful information regarding the adsorption process as well.
We absolutely agree with the reviewer; out manuscript needed an updating of references: Indeed, we have enlarged the state of art, and we have included new references; instead of including those references suggested by the reviewer, we have found others, which specifically refer to our adsorbates: fluoxetine and nicotinic acid. In particular, the following references have been added to the manuscript:
[15] Health at a glance 2018: State of health in the eu cycle. European Comission Report. 2018.
[16] Li, Y.; Zhang, S.; Zhang, W., Xiong, W.; Ye, Q.; Hou, X.; Wang, C. Wang, P. Life cycle assessment of advanced wastewater treatment processes: Involving 126 pharmaceuticals and personal care products in life cycle inventory. J Env Manag 2019, 238, 442–450.
[18] Sousa-Moura, D.; E. Y. Matsubara et al. CNTs coated charcoal as a hybrid composite material: Adsorption of fluoxetine probeby zebrafish embryos and its potential for environmental remediation. Chemosphere, 2019, 230, 369-376.
[19] Silva, B.; Martins, M.; Rosca, M.; Rocha, V.; Lagoa, A.; Nevesa, I. C.; Tavares, T. Waste-based biosorbents as cost-effective alternatives to commercial adsorbents for the retention of fluoxetine from water. Sep Purif 2020, 235, 116-139.
[20] Bruckert, E.; Labreuche, J.; Amarenco, P. Meta-analysis of the effect of nicotinic acid alone or in combination on cardiovascular events and atherosclerosis. Atherosclerosis 2010, 210(2), 353–361.
[21] Datta, D.; Sah, S.; Rawat, N.; Kumar, R. Application of Magnetically Activated Carbon for the Separation of Nicotinic Acid from Aqueous Solution. J Chem Eng Data 2017, 62, 712−719.
- The conclusion seems somewhat lengthy and mild concentration is suggested.
We have followed the reviewer suggestion and rewritten the conclusions, decreasing the number of words from 257 to 192 (more than 75%), as follows:
“Activation of biomass-based hydrochars using air and carbon dioxide can provide adsorbents with diverse porosity and surface acidity properties that can be used to remove fluoxetine and nicotinic acid, although the most suitable option was strongly dependent on the adsorbate.
It was found that adsorption process was more influenced by the surface chemistry than by the porous structure of the carbons. Fluoxetine showed removal results when air activated carbon were used, whilst in the case of nicotinic acid the maximum adsorption capacity was achieved for samples produced by carbon dioxide. This behavior can be justified by the non-dispersive interaction, chemical bonding and electrostatic interactions between the activated carbons surface and the adsorbate species.
Among the different adsorption models used (Langmuir, Freundlich, Redlich-Peterson and Sips) the first one predicted more accurately liquid the adsorption isotherms. Values of Q0 up to 44.1 and 91.9 mg g-1 were obtained for fluoxetine and nicotinic acid adsorption which were, respectively, similar or superior to those found in the literature for other commercial and home-made adsorbent. The convenience of using hydrocarbonization followed by activation is noteworthy because of its simplicity, low energetic and economical cost and environmental advantages.”

Reviewer 3 Report
The manuscript is well written and presented some results on hydrothermal carbonization and activation of différent agrowastes. The performance of the hydrochars obtained as sorbents is evaluated for 2 organic molecules. it is a good work but still some details has to be clarified:
1) How the original shell have been selected?
2) Some information has to be given about the adsorption performance of the original sample and the not activated hydrochar.
3) PTZ and surface area have to be evaluated and mentioned without activation to illustrate the advantage of the activation
4) the activated biochars has to be characterized by SEM or TEM. The chemical composition of the surface has also to be characterized by FTIR, XPS, or EDS spectroscopy
5) Why the surface area of sunflowed-dervied biochar is less modified by activation than the other materials?
6) Figure 1 has to be corrected: there is 2 black curves and cross point has to be black to fit with legends.
7) Figure 1 and 2 has to present all the datas from 10 to 200 mg/L as it is described in the experimental part
8)The unit of Kf has to be homogenize between table 2 and experimental part
9) how many replicate are made for the adsorption experiment? T It sould be indicated
10) The adsorption rate seems to be interesting but there is no comparison with already published data form such material.It sould be added
11) Each molecule were tested separately but to be suitable for industrial application, this process has to be tested with different molecules to evaluate its selectivity.
13) Electrostatic interaction seems to be important in the adsorption process. So, the authors should evaluate the influence of the pH of the solution on the adsorption rate.
Author Response
Answers to reviewer 3,
The first thing we would like to do is to thank the reviewer for his/her time and effort to improve our manuscript; specially in the frame of this strange situation we are living, he/she found the time to devote his/her knowledge to this work, and we appreciate it very much. We have answered the suggestions point by point, as we explain below:
1) How the original shell have been selected?
We understand that no information on the origin of the precursors, and also about how they were received and treated before use. In order to clarify these issues, we have added the following information to the manuscript:
“The adsorbents used in this study were all made by HTC of three biomass materials (walnut shell, sunflower stem, and olive stone), followed by physical activation. The precursors were chosen because of their abundancy in the Iberian Peninsula, and were supplied by local cooperatives from Extremadura (walnut shell and olive stone) and Alentejo (sunflower stem) areas, in Spain and Portugal, respectively. All the three materials were used “as received” (without any washing), and then subjected to crushing and grinding to a particle size of 1-2 mm, as described in previous papers [2].”
2) Some information has to be given about the adsorption performance of the original sample and the not activated hydrochar.
We did not subject to adsorption performance test to the precursor nor to the non-activated HCs in this paper, because we have tested as part of previous research other biomass samples and hydrochars. One example can be found at our previous work in:
- Development and characterization of activated hydrochars from orange peels as potential adsorbents for emerging organic contaminants (M. E. Fernández, 2015)
In general, these materials do not have a developed porous network and because of that, it is necessary to perform chemical or chemical activation. In addition, both biomass and specially HC are associated to tarnish effects of the water, and in consequent the measurement of the target species in solution is difficult, because some signals (corresponding to organic compounds from the tarnish of the water) is overlapped. The adsorption isotherms in solution show that due to these effects, the adsorption is hindered, and the application of the adsorption models is not possible.
If, after this explanation, the reviewer considers that we should test the original biomass materials and the HCs, it would involve additional experimental work that in addition we are not in conditions of doing (the University is closed due to the confinement). We hope the reviewer and the editor accept out point of view.
3) PTZ and surface area have to be evaluated and mentioned without activation to illustrate the advantage of the activation
We understand the suggestion made by the reviewer. We had this characterization from our previous studies and we have included it in the manuscript, in the presentation of the textural properties of the adsorbents (page 3):
“It is interesting to recall on the significant increase on SBET that the activation brought out, in relation to the original HCs, for which it had values of 31, 27 and 22 m2/g-1, respectively. Besides, all the three HCs were very acidic in nature before activation processes (4.12, 4.01 and 3.95, following the same sequence).”
4) the activated biochars has to be characterized by SEM or TEM. The chemical composition of the surface has also to be characterized by FTIR, XPS, or EDS spectroscopy
The activated HCs have already been characterized by SEM and FT-IR; as we explain in the paper, this characterization has been previously included in a paper (ref. 11 of the manuscript).
5) Why the surface area of sunflowed-dervied biochar is less modified by activation than the other materials?
We guess the reviewer means that the HCs from sunflower are less influenced by the type of activating agent used that the other two biomass precursors. In fact, from our point of view, in this study, the air activated samples did not develop all their potential porosity, in the case of olive stone and walnut shell, probably because of external burning during air activation, due to the high reactivity of the materials. According to the bibliography on the porosity development potential of the same walnut shell and olive stone by means of traditional physical activation, and we found that they can be much higher.
So, the origin of the better behavior of sunflower is related to the fact that walnut shell and olive stones HCs were burnt in some extent. This information is consistent with the greater burn-off values, close to 70% for these two materials, whereas it was 40% for sunflower, and also to the greater contribution of external area in the case of the former materials. We have added some information about this to the manuscript, as follows:
“On the other hand, it is also interesting to notice that, while CO2 processes resulted in a similar porosity development (in all cases microporous carbons with SBET values close to 400 m2g-1, air behaved better with sunflower stem. Based on previous research with the same precursors on traditional air activation processes, the authors believe that, in fact, walnut shell and olive stone did not develop all their potential porosity [22,23]. In this work, the results of high burn-off for these two materials after air activation and also the bigger contribution of mesoporosity suggest that some external burning might have taken place.”
Since this change involved the addition of new references, we renumbered all them along the manuscript.
6) Figure 1 has to be corrected: there is 2 black curves and cross point has to be black to fit with legends.
We absolutely agree with the suggestion of the reviewer and have modified Figure 1, in accordance with his/her suggestion. We have taken this opportunity to improve the series visibility by modifying their color and we have also adapted the figure font to the one of the text (palatino linotype).
7) Figure 1 and 3 have to present all the datas from 10 to 200 mg/L as it is described in the experimental part
The experimental part of the manuscript describes the conditions under we made the adsorption isotherms. The range 100-200 mg/L describes the initial adsorbate concentration (before contact with activated carbons). After equilibrium, this concentration decreases, giving the values shown in Figure 1, ranging between 0-40 and 0-90 mg/L, for fluoxetine and nicotinic acid, respectively. We think that making larger the axes does not make sense, because all values obtained after adsorption fit within the ranges described.
8)The unit of Kf has to be homogenize between table 2 and experimental part
We have corrected this issue and homogeinized the unit of Freundlich constant: (mg g-1)(L mg-1)1/n
9) how many replicates are made for the adsorption experiment? It should be indicated
Each adsorbance test was made in triplicate; we have added the corresponding information to the manuscript:
“Each measurement was made in triplicate, and the absorbance used to calculate the adsorbate concentration corresponds to the average value”
10) The adsorption rate seems to be interesting but there is no comparison with already published data form such material.It should be added.
We do not understand what the reviewer means by adsorption rate; we did not study the kinetics of adsorption, but the equilibrium adsorption. By making preliminary tests, we made sure that after 48 h, the equilibrium had been reached, but we did not study how the concentration of the adsorbents changed with time.
11) Each molecule were tested separately but to be suitable for industrial application, this process has to be tested with different molecules to evaluate its selectivity.
We really understand the point of the reviewer at this point. We plan to make such experiments in the future, in order to guarantee the real good performance of the adsorbents. The first step, however, had to be to test them separately, to know what type of structural properties and surface chemistry was more convenient, and to understand each adsorption mechanisms, without interferences. We thank the reviewer by this suggestion, that we will sure follow in our next works. We have added a sentence about this intention in the manuscript:
As final remark, we would like to emphasize how, once the preparation of the adsorbents is optimized (in terms of control of porous structure and surface functionality), testing them is essential to investigate their performance towards each adsorbate. Two organic drugs, such as fluoxetine and nicotinic acid, can show enhanced affinity towards one adsorbent that, at first glance, could seem less interesting. For this reason, taking the specific surface or the pore volumes is not enough, because the pore size distribution and the amount and type of functional groups can be decisive to favor adsorption or repulsion processes. The authors would also add that real adsorption in wastewater plants takes place with multiple adsorbates, and, in this line, future steps will be devoted to study the performance of these type of materials in multisorption systems, mimicking the conditions of real cases.
13) Electrostatic interaction seems to be important in the adsorption process. So, the authors should evaluate the influence of the pH of the solution on the adsorption rate.
We see the point of the reviewer. It would be very interesting to measure the adsorbate concentration with time, and also how the solution pH is modified during the process. Unfortunately, as we have stated previously, in this work we did not focus on kinetics but on equilibrium results. We thank the reviewer about the suggestion, and we will include it in our next pieces of research.
Round 2
Reviewer 3 Report
The authors answered to the reviewer comment. Now, the article is suitable for publication